



*Technical Note:* **Monte-Carlo genetic algorithm (MCGA) for**
**model analysis of multiphase chemical kinetics to determine**
**transport and reaction rate coefficients using multiple**
**experimental data sets**
**Thomas Berkemeier[1,*], Markus Ammann[2], Ulrich K. Krieger[3], Thomas Peter[3], Peter**
**Spichtinger[4], Ulrich Pöschl[1], Manabu Shiraiwa[1,5] and Andrew J. Huisman[6]**
[1] Max Planck Institute for Chemistry, Multiphase Chemistry Department, 55128, Mainz,
Germany
[2] Paul Scherrer Institute, Laboratory of Environmental Chemistry, 5232, Villigen, Switzerland
[3] ETH Zurich, Institute for Atmospheric and Climate Science, 8092, Zurich, Switzerland
[4] Johannes Gutenberg University, Institute for Atmospheric Physics, 55128, Mainz, Germany
[5] University of California Irvine, Department of Chemistry, 92697, Irvine, CA, USA
[6] Union College, Department of Chemistry, 12308, Schenectady, NY USA
*now at: Georgia Institute of Technology, School of Chemical and Biomolecular Engineering,
30320, Atlanta, GA, USA
**Corresponding Authors:** T. Berkemeier (thomas.berkemeier@chbe.gatech.edu) and A. J.
Huisman (huismana@union.edu)


## 20    Abstract

We present a Monte-Carlo Genetic Algorithm (MCGA) for efficient, automated and unbiased
global optimization of model input parameters by simultaneous fitting to multiple experimental
data sets. The algorithm was developed to address the inverse modelling problems associated with
fitting large sets of model input parameters encountered in state-of-the-art kinetic models for
heterogeneous and multiphase atmospheric chemistry. The MCGA approach utilizes a sequence
of optimization methods to find the solution of an optimization problem and to explore the space
of solutions with similar model output. It addresses a problem inherent to complex models whose
extensive input parameter sets might not be uniquely determined from limited input data. Such
ambiguity in the derived parameter values can be reliably detected using this new set of tools. The
MCGA algorithm has been used successfully to constrain parameters such as reaction rate
coefficients, diffusion coefficients and Henry's law solubility coefficients in kinetic models of gas
uptake and chemical transformation of aerosol particles as well as multiphase chemistry at the
atmosphere-biosphere interface. It should be portable to any numerical model with similar
computational expense and extent of the fitting parameter space.

## 36    1.  Introduction

Atmospheric aerosols play a key role in climate, air quality and public health. Heterogeneous
reactions and multiphase processes alter the physical and chemical properties of organic aerosol
particles, but the effects of these reactions are not fully elucidated (e.g. Finlayson-Pitts,
2009;George and Abbatt, 2010;Abbatt et al., 2012;Pöschl and Shiraiwa, 2015). While multiphase
chemistry in aerosols and clouds can be described by a sequence of well-understood physical and



chemical elementary processes in kinetic models (Hanson et al., 1994;Pöschl et al., 2007;George
and Abbatt, 2010), the deduction of parameters or rate coefficients of the individual elementary
processes is severely complicated by the inherent coupling of chemical reactions and mass
transport processes (Kolb et al., 2010;Berkemeier et al., 2013;Shiraiwa et al., 2014).
Heterogeneous chemical reactions on aerosol particles are traditionally described using so-called
"resistor" models, which represent parallel and sequential physical or chemical processes in
analogy to electrical circuits. These models have typically been used to derive analytical
expressions for simplified limiting cases (e.g. Hanson et al., 1994;Worsnop et al., 2002;Hearn et
al., 2005). Recently, numerical models have been developed that allow a more complete
consideration of the time- and depth-resolved chemical and physical behaviour of aerosol particles,
leading to a better understanding of these reaction systems, especially under conditions where the
steady-state assumptions underlying the resistor models are not valid (Smith et al., 2003;Pöschl et
al., 2007;Steimer et al., 2015;Berkemeier et al., 2016). Kinetic multi-layer models describe single
particles or thin films by division into compartments such as near-surface gas phase, surface and
particle bulk, and further subdivision of the particle bulk into thin layers to achieve depth-
resolution. Specific models provide a focus on chemistry such as KM-SUB (Shiraiwa et al., 2010),
on gas-particle partitioning such as KM-GAP (Shiraiwa et al., 2012) and ADCHAM (Roldin et al.,
2014), or on water diffusion such as the ETH Diffusion Model (Zobrist et al., 2011). For simplicity,
throughout this manuscript we refer to a "kinetic model" as any computational model that is used
to simulate a system's behaviour and to "input parameters" as any parameters (thermodynamic,
kinetic, or physical) that need to be optimized so that the kinetic model accurately represents
experimental data.



Ideally, fitting a kinetic model to experimental data would return all chemical and physical
parameters necessary to understand the importance of the processes at work and to predict the
outcome of future experiments, even if conducted under experimental conditions not part of the
training data set, i.e. all experimental data used during the fitting process. However, kinetic models
often require a multitude of input parameters, some of which are not constrained well
experimentally or are merely effective parameters combining a sequence of inherently coupled
processes. In general, two main difficulties arise when optimizing complex models to experimental
data:
(1) The optimization hyper surface is often non-convex, i.e., it will not have only a single minimum
due to interactions between non-orthogonal input parameters and/or scatter in the experimental
data. Hence, steepest descent methods fail since they get trapped easily in local minima. Brute-
force or exhaustive searches, where an $n$-dimensional grid is applied to the input parameter space
and the fit quality evaluated for every grid point in all $n$ dimensions, are often not computationally
feasible.
(2) If too little or too similar experimental data is used during the fitting process or input
parameters are allowed to move in a large range, the optimization problem can be underdetermined
(ill-defined) and multiple solutions may exist. In this case, even though a good agreement between
model output and training data set is obtained, it is likely that only the model input parameters
corresponding to the most limiting processes will be physically meaningful. Extrapolation of the
model outside its training range can then lead to strong discrepancies between modelled and
measured data.



Hence, sophisticated optimization methods are needed, which quickly and reliably determine the
model input parameters that lead to the best correlation between model and experimental data.
Furthermore, experiments must be conducted by covering broad ranges of experimental conditions
to achieve that the observables are controlled by (a) as many model input parameters as possible
across all experimental conditions, but (b) by as few model input parameters as possible for a
specific experimental condition (i.e. limiting cases). Note here that technical limitations or
transient behaviour of a reaction system may not allow probing the entirety of the parameter space
in the required breath.

## 2.  Monte-Carlo Genetic Algorithm (MCGA)

In many modelling applications, methods are needed that reliably find the optimum in non-convex
optimization problems and detect underdetermined optimization problems. Global optimization
methods have been subject of extensive research in the past (Arora et al., 1995) and provide means
of approximating non-convex optimization problems without premature convergence to local
optima. Examples for these methods are simulated annealing methods and evolutionary
algorithms. In atmospheric chemistry, simple optimization techniques are commonly used to
determine kinetic parameters by fitting of rate equations to experimental data sets, but to our
knowledge no global optimization technique diligently designed for the determination of
atmospheric reaction rate coefficients from multiple data sets was described thus far. For cloud-
aerosol interaction models, inverse modeling techniques using evolutionary algorithms as global
optimization technique in Monte-Carlo Markov Chain (MCMC) algorithms were developed
previously to determine parametric uncertainties (Partridge et al., 2012;Lowe et al., 2016). Global

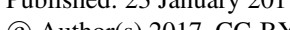


optimization was also used to calculate thermodynamic equilibria for phase separation of aqueous
multicomponent solutions (Zuend and Seinfeld, 2013).
In this study, we present the Monte-Carlo Genetic Algorithm (MCGA), a method combining direct
Monte-Carlo sampling with a genetic algorithm as heuristic global optimization method that
approximates the global optimum for input parameter sets of computational models. Repeated
execution of the search algorithm can be used to test for uniqueness or to provide statistical bounds
on the model input parameters. The MCGA algorithm utilizes a two-step approach to find minima
on non-convex hyper surfaces. First, a Monte-Carlo (MC) sampling is performed in the large space
of possible model input parameters to narrow down the possible solution to smaller areas of
interest. The parameter sets are evaluated using a goodness-of-fit expression of the user's choice,
such as the root-mean-square (RMS) error. In the examples presented here, the RMS error or
logarithmic RMS error was used. When multiple datasets were fitted, a weighting factor was
introduced to prevent bias due to the number of data points in different experimental datasets. An
additional optional weighting factor allows the user to assign priority to experimental data with
lower statistical error or scatter. The parameter sets for the MC sampling are generated randomly
from a distribution of the model input parameters. Each parameter was sampled using a
logarithmically spaced distribution of values to provide uniform sampling over the large ranges
most input parameters can possibly adopt. Note that, depending on the problem, different
distributions and sampling strategies (e.g. Latin hypercube sampling) could be applied.
The genetic algorithm (GA) uses survival of the fittest to optimize an ensemble (the *population)*
of parameter sets (the *individuals*) over several iterations (the *generations*). Processes known from
natural evolution such as survival, recombination, mutation and migration are mimicked to
optimize a population. The initial population is formed by the parameter sets with the best





goodness-of-fit obtained in the MC sampling step. An equal number of random parameter sets are
added to ensure diversity within the pool of parameter sets and counteract sampling bias from
shallow local minima (Fig. 1).
During execution of the GA, a number of model input parameter sets with the highest correlation
between model output and experimental data (goodness-of-fit) are directly transferred into the next
generation by the survival mechanism (the *parents*). The remaining population is recombined to
generate new combinations of parameters from the existing sets, forming the *children* for the next
generation. To further ensure genetic variability, a mutation scheme alters parameters in a
stochastic manner. For the same purpose, the MCGA code allows for optional reseeding of a
population during optimization, i.e. replacing random individuals in the population with random
or pre-sampled individuals from the preceding MC run, and for optional migration between
different populations if multiple subpopulations are being evolved concurrently. Collectively,
these mechanisms enable the MCGA to overcome local minima, a crucial feature of a global
optimization method. Iteration of these steps eventually results in a homogeneous, optimized
population and the common parameter set is taken as result. In lieu of reseeding and migration,
MCGA can be run multiple times to generate a set of representative solutions, which has been the
default approach in previous applications of MCGA (cf. Sect. 4). With only few (~5-10)
repetitions, this procedure allows the user to assure full convergence to the global optimum. In
addition, the random sampling of optimization space between different executions of MCGA will
generate statistical bounds on the parameters if a sufficiently large number of repetitions is
computationally feasible.





In this study we used the genetic algorithm provided by MathWorks® (Matlab® Global
Optimization Toolbox) and developed a routine for parallel computation on computer clusters. In
a typical setting, the MC step and GA step of the optimization occupied an approximately equal
amount of computation time. Figure 2 describes the implementation of the parallel MCGA
optimization method. The $N$ parallel threads share common populations of parameter sets that are
iteratively optimized by extracting a subset of parameter sets and performing the genetic algorithm
on this subset. Once a sub-evaluation of the genetic algorithm has finished, the parameter sets are
mixed into the population, and after randomization, a different subset of parameter sets is extracted
and their optimization is immediately continued. Since the parallel threads will run
asynchronously, a fraction of individuals must remain in the population to be mixed with, to enable
continuous operation without waiting times.

### 163     3. Implications for modelling and measuring chemical kinetics

Although models may possess a multitude of kinetic and thermodynamic input parameters that
represent the many possible sequential and/or concurrent processes occurring in the system, their
behaviour is often driven by only a single or at most a few processes at a certain point in time. In
chemical kinetics, the behaviour of the system can often be characterized by a kinetic regime,
which may change during the course of the reaction and with experimental conditions (Berkemeier
et al., 2013). If a set of model input parameters can be uniquely determined (by MCGA or another
means) and results in a high-fidelity fit of model output to experimental data, the parameters then
would be regarded as correct within the approximations of the underlying model and uncertainties
of the experimental data. This is a convenient way to assimilate data from multiple previous





studies; data sets can be weighted to reflect confidence in their results, and the final range of
accepted parameters then represents a consensus from the fitted data. However, it may not always
be possible to fully constrain the input parameters, even using multiple experimental datasets. In
general, there are two reasons that a model input parameter can remain unconstrained after
optimization:
(i)      the parameter is non-influential, or
(ii)     the parameter is inherently coupled to another one, forming a non-orthogonal parameter

180             pair under all experimental conditions.

Fig. 3 illustrates both cases in an example taken from atmospheric multiphase chemistry, using the
benchmark system of ozone + oleic acid and data adopted from Hearn *et al.* (2005). The original
data was converted from ozone exposure to a time series using an ozone concentration of $2.76 \times 10^{15}$
$cm^{-3}$. The MCGA algorithm was executed under a constrained parameter set, in which only
desorption lifetime and surface reaction rate coefficient were allowed to vary. In this scenario,
repeated execution of MCGA returned multiple solutions, for which the model output had nearly
equivalent goodness-of-fit with only slight variance between them (Fig. 3A). In stark contrast to
the uniform correlation between model output and experimental data, Fig. 3B shows the high
variance within the model parameters yielding these solutions (red markers) which scatter across
a narrow valley of the optimization hypersurface (contour lines). In the upper portion of the figure,
i.e. above a desorption lifetime of $10^{-4}$ s, a vertical relationship between both parameters indicates
that the desorption lifetime is a non-influential parameter and can take on any value in this interval,
corresponding to case (i) above. In the lower portion of the figure, i.e. below a desorption lifetime
of the diagonal relationship indicates that an increase in one parameter can be compensated with a
decrease in the other parameter and both form a non-orthogonal pair, corresponding to case (ii)





above. For comparison, Figs. 3C and 3D show examples of optimization hypersurfaces from
Berkemeier et al. (2016), who studied multiphase ozonolysis of shikimic acid and investigated the
existence of non-orthogonal parameter pairs by varying optimized parameters ($\lambda_i$) by a factor $f(\lambda_i)$
to depict the total residual as a 2D contour map. Fig. 3C shows that the Henry's law coefficient for
ozone ($H_{cp,O3}$) and the product of the bulk reaction rate coefficient ($k_{BR}$) with the bulk diffusivity
of ozone ($D_{b,O3}$) and the bulk-to-surface transport coefficient of ozone ($k_{bs,O3}$) are fully non-
orthogonal. Figure 3D shows a single, well-defined optimum parameter set for the effective
molecular cross section of ozone ($\sigma_{O3}$) and the desorption lifetime of ozone ($\tau_{d,O3}$), indicating that
these parameters are fully orthogonal for the experimental data fit in that study.
The prerequisite of a successful optimization is to fit a sufficiently broad experimental data set so
that a unique and accurate set of fitting parameters is obtained. Thus, both of the above conditions
must be avoided. This may be achieved by including additional experimental data, especially from
a different experimental technique or over a different timescale so that the system might sample
another limiting behaviour. In the data given in Fig. 3 above, for example, measuring full time
series at different oxidant concentrations may help to constrain the oxidant's desorption lifetime.
However, if a model has too many free parameters (or especially parameters that are not well-
constrained by experimental data), it may be necessary to reduce the model complexity or fix some
of the parameters. We therefore recommend using data sets obtained from a range of different
experimental techniques to ensure this variability if they are available, and using models with as
few free parameters as possible.
In the example above, it was possible to use brute-force sampling to determine the true
optimization hypersurface (contour lines) for comparison to the MCGA results. Of course, in
typical applications, the number and range of input parameters makes such a search prohibitive.



The computational feasibility of an optimization depends crucially on the size of the input
parameter space, i.e. number and possible range of all parameters. Using an unreasonably large
range for input parameters increases the possibility of finding non-physical solutions that fit the
experimental data. The input parameter space can be reduced based on *a priori* knowledge from
laboratory experiments and theoretical calculations. Parameters can be narrowed down by
laboratory experiments (e.g. bulk experiments for derivation of trace gas solubility), by physics
(e.g. the upper limit of the accommodation coefficient at unity), or by simulations (e.g., molecular
dynamics simulations to estimate the surface accommodation coefficient and desorption lifetime
as in Vieceli et al. (2005) and Julin et al. (2013)). Note that in the example given in Fig. 3b, the
two parameters were not truly independent, so that constraining either model parameter from *a*
*priori* information would constrain the other parameter. In multi-parameter optimizations, where
many such dependencies might exist, this can lead to a significant reduction in solution space.

## 4. Application of MCGA in atmospheric multiphase chemistry

The MCGA algorithm has been applied previously to chemical reaction systems of atmospheric
relevance (Table 1). The essential parameters we use to describe an atmospheric multiphase
chemical kinetic system of reactive trace gases X and bulk material Y include chemical reaction
rate coefficients at the surface ($k_{SLR}$) and in the bulk ($k_{BR}$) of aerosol particles; bulk diffusion
coefficients of reactive trace gases ($D_X$) and the bulk matrix ($D_Y$); accommodation coefficients
($\alpha_{s,X}$) and desorption lifetimes ($\tau_{d,X}$) of trace gases to the particle surface to determine transient and
equilibrium adsorption behavior; and equilibrium constants for the solubility of reactive trace





gases ($K_{sol,cc,X}$), typically expressed in terms of Henry's law coefficients ($H_{cp,X}$) (Pöschl et al.,
2007;Ammann and Pöschl, 2007;Shiraiwa et al., 2010;Berkemeier et al., 2013).
In its first application the MCGA algorithm was used to fit individual data sets of the decay of
oleic acid upon ozonolysis (Berkemeier et al., 2013), highlighting the need of fitting to multiple
experimental data sets to constrain kinetic parameters. This was done in further studies that
investigated gas uptake to (semi-)solid organic material in coated-wall flow-tube reactors (Arangio
et al., 2015;Berkemeier et al., 2016), ozone-induced protein oligomerization in bulk solutions
(Kampf et al., 2015), viscosity change upon alkene ozonolysis as measured with fluorescence
microscopy (Hosny et al., 2016) as well as the redox-cycling reactions in the human lung lining
fluid (Lakey et al., 2016). In each of these studies, a large set of model input parameters was
optimized to several kinetic data sets to constrain the input parameter space. In the following, we
review results previously obtained by the MCGA algorithm to demonstrate its utility in
determining kinetic parameters, assimilating large datasets, and detecting ill-defined problems.
In Berkemeier et al. (2016), 11 parameters were varied simultaneously to fit the ozone uptake to
shikimic acid films over many hours, under 12 distinct experimental conditions, and using a single
set of kinetic parameters (Fig. 4). The model was found to accurately describe the humidity- and
concentration-dependence of ozone uptake and a high correlation between model output and
experimental data was achieved. During optimization, a subset of six parameters, including
diffusivity coefficients and trace gas solubility, was allowed to increase or decrease monotonically
over 6 steps in relative humidity, resulting in a total of 41 optimized parameter values. Despite this
large number of optimization parameters, a well-constrained parameter set could be obtained due
to the large depth in training data and by applying *a priori* information.



In another study investigating the oxidation of biomass burning tracers with hydroxyl radicals
(Arangio et al., 2015), repeated execution of MCGA revealed a remaining uncertainty in the kinetic
parameters obtained from optimization to the two experimental data sets (Fig. 5). While some
parameters could be narrowly constrained (diffusion coefficient of the organic matrix, $D_{org}$), others
were subject to larger uncertainties (surface layer reaction rate constant $k_{SLR}$, desorption lifetime
$\tau_d$). Note that while these parameters seem almost unconstrained in Fig. 5, this uncertainty is due
to the presence of non-orthogonal parameter pairs. As detailed in Fig. 3 and in Arangio et al.
(2015), only specific combinations of the non-orthogonal parameters will lead to agreement
between model and experiment. This knowledge can be used to constrain these parameters in
further experiments.
**5.  Conclusions**
The MCGA algorithm addresses the problem of extracting physical and chemical parameters from
experimental data. The algorithm allows the user to assimilate multiple datasets and its random
sampling approach reduces the bias which may arise in more user-directed optimization methods.
Unlike simple gradient-based optimization methods, MCGA can thus be used as a statistical tool
that not only detects unconstrained parameters, but also finds dependencies between unconstrained
parameters. The results can be applied in process models and may serve to direct future
experimental studies, e.g. to drive a reaction system into regimes in which the remaining
unconstrained parameters have high sensitivity. MCGA could also be used to constrain chemical
reaction systems in the post-analysis of field and laboratory studies: starting with a large set of
model input parameters (i.e. chemical reactions, physical processes), data from various
measurement campaigns could be combined, reconciled and in a further step used to reduce the
number of model input parameters to the key processes necessary to describe all measurement

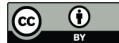


data. MCGA may be a powerful and useful tool to constrain kinetic parameters and reaction rate
coefficients in models that study the formation of secondary organic aerosol in reaction chambers
(Chan et al., 2007;Shiraiwa et al., 2013;Cappa et al., 2013;Riedel et al., 2016). It could be suitable
for fine-tuning of reaction rates in large reaction mechanisms of atmospheric chemistry, such as
the Master Chemical Mechanism (MCM; Jenkin et al., 1997;Saunders et al., 2003), the Gas-
Aerosol Model for Mechanism Analysis (GAMMA; McNeill et al., 2012) or the Chemical
Aqueous Phase Radical Mechanism (CAPRAM; Herrmann et al., 1999). Multiple experimental
data sets from a broad range of techniques could be used with the algorithm to narrow down
difficult-to-measure reaction rate coefficients, provide uncertainty estimates and reconcile
experiments across different research groups and facilities.
**Acknowledgements**
T. Berkemeier was supported by the Max Planck Graduate Center with the Johannes Gutenberg-
Universität Mainz (MPGC). A. J. Huisman was supported by the United States National Science
Foundation under award no. IRFP 1006117 and by ETH Zürich. Any opinions, findings, and
conclusions or recommendations expressed in this material are those of the authors and do not
necessarily reflect the views of the US National Science Foundation. We gratefully acknowledge
G. D. Smith for providing published data in tabulated form. The authors like to thank A. Pozzer,
C. Pfrang and C. Marcolli for stimulating discussions and support.
**Table 1.** Previous studies applying the MCGA algorithm.

| Study | Reaction system |
|---|---|
| Berkemeier et al. (2013) | oleic acid + $O_3$ |
| Arangio et al. (2015) | levoglucosan and abietic acid + OH |
| Kampf et al. (2015) | protein + $O_3$ |
| Hosny et al. (2016) | oleic acid + $O_3$ |
| Berkemeier et al. (2016) | shikimic acid + $O_3$ |
| Tong et al. (2016) | OH formation by SOA decomposition in water |
| Lakey et al. (2016) | reactive oxygen species and PM2.5 in lung lining fluid |




# Monte-Carlo Genetic Algorithm

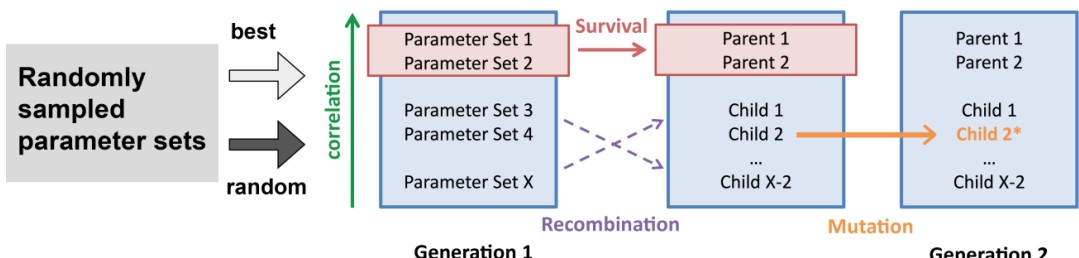

**Figure 1.** Schematic representation of the MCGA optimization method consisting of a Monte-Carlo sampling, which feeds into a genetic algorithm. Through survival, recombination and mutation steps, ensembles of model input parameter sets are iteratively improved over several generations until a sufficient correlation to the experimental data is obtained.





310

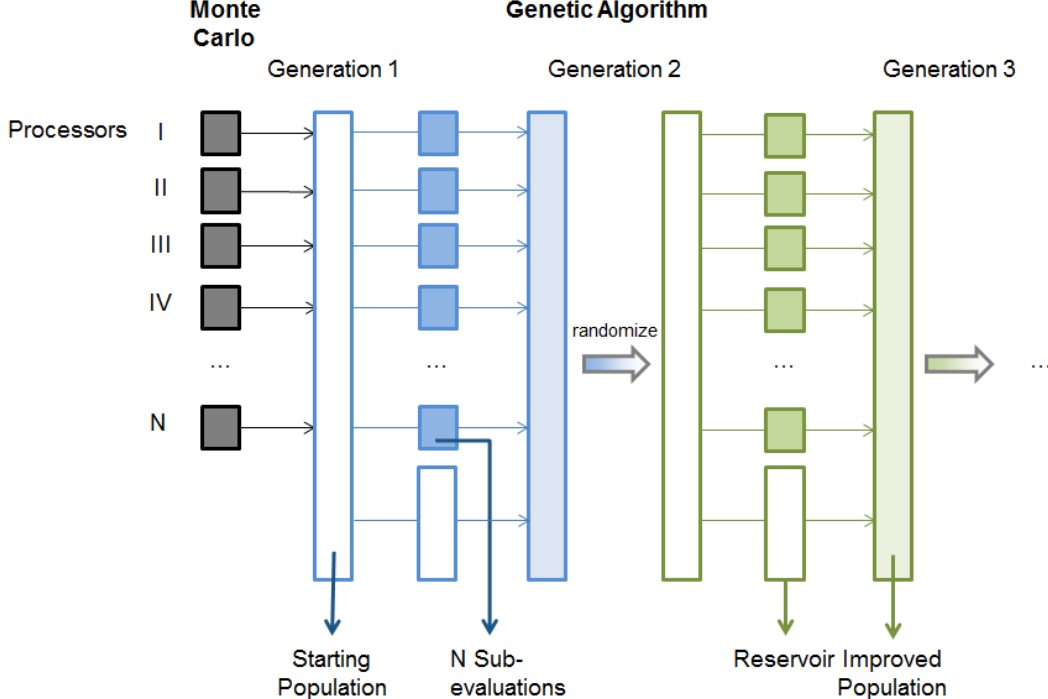

**Figure 2.** Schematic visualization of the parallelized MCGA optimization method. The Monte Carlo step

is performed independently on $N$ processors and the best fitting parameter sets are fed along with random

parameter sets into the starting population. During the genetic algorithm step, each processor extracts a

number of parameter sets from the collective pool and performs a sub-evaluation of the genetic algorithm

on these parameter sets. After completion, the optimized parameter sets are fed back into the pool, which

always contains a non-zero number of parameter sets as reservoir. After randomization, a different

combination of parameter sets is extracted and the process repeated.

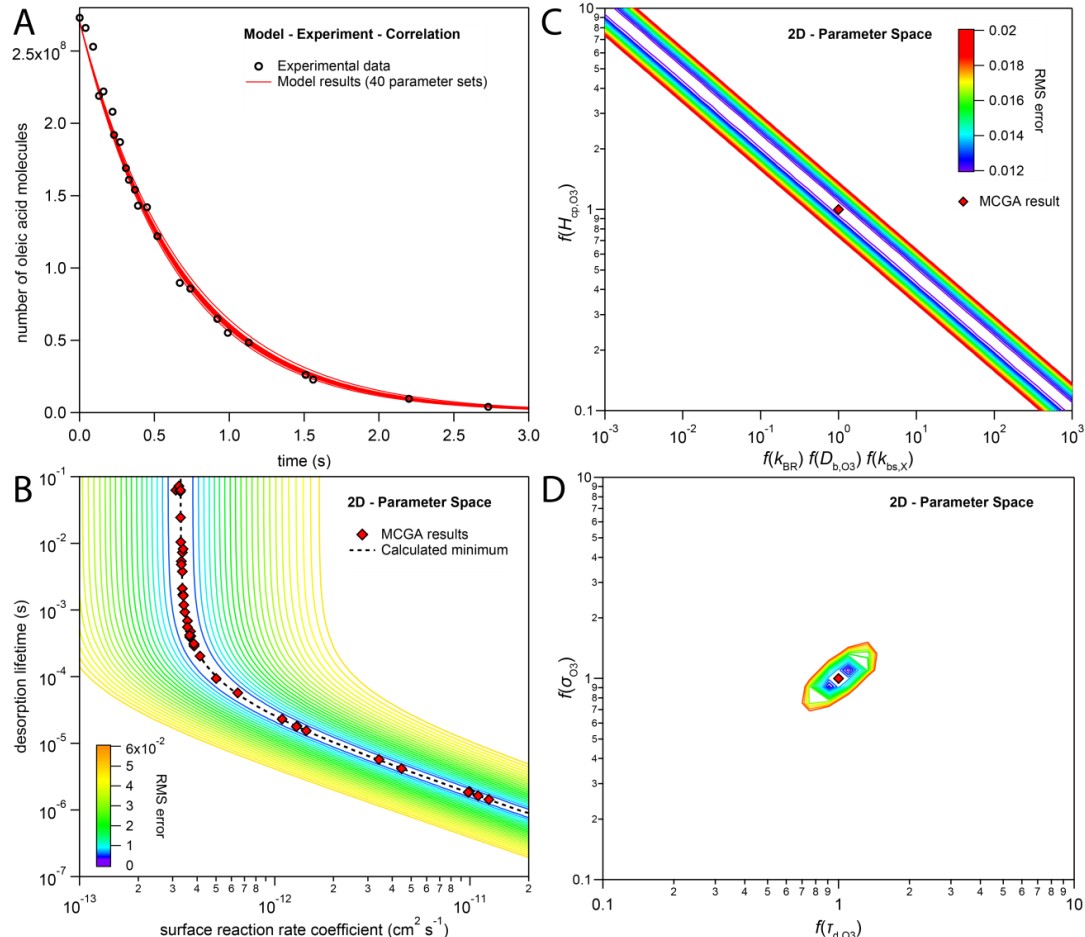

318

**Figure 3.** (A) Results from repeatedly fitting a kinetic model to a single experimental decay curve (adopted

from Hearn et al., 2005). MCGA was used to optimize two model parameters, a surface reaction rate

coefficient and the desorption lifetime of the gas phase oxidant. All other model parameters remained fixed.

(B) Visualization of MCGA algorithm's findings on the 2-dimensional optimization hypersurface. The

hypersurface (contour lines represent the root mean square deviances) exhibits no unique minimum due to

insufficiently broad experimental data and optimization results (red diamonds) scatter along the extended

minimum (black dashed line). (C) and (D) show exemplary optimization hypersurfaces with two parameters



showing an elongated (C) or a distinct minimum (D). Panels C and D are reproduced from Berkemeier et
al. (2016) with permission from the PCCP Owner Societies.

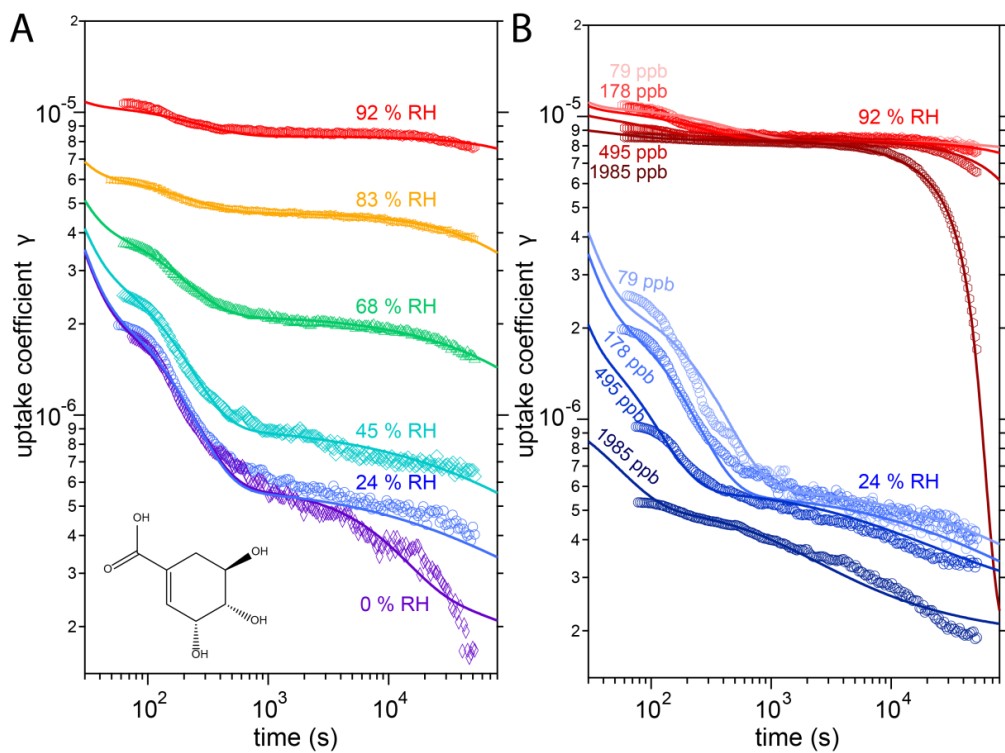

**Figure 4.** Observed (markers) and modelled (lines) uptake coefficients of ozone onto a thin film of shikimic

acid as a function of exposure time. (A) Uptake coefficients at 178 ppb ozone gas phase concentration $[O_3]_g$
at different relative humidities of 0, 24, 45, 68, 83, and 92%. The structural formula of shikimic acid is
displayed in the left bottom corner. (B) Uptake coefficients at 24% RH (blue solid lines) and 92% RH (red
solid lines) with different $[O_3]_g$ of 79, 178, 495, and 1985 ppb. Reproduced from Berkemeier et al. (2016)
with permission from the PCCP Owner Societies.



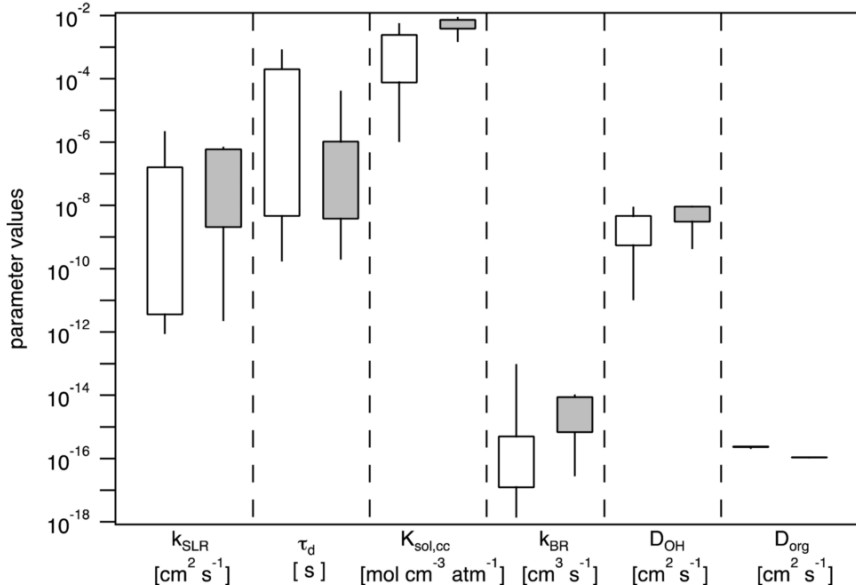


**Figure 5.** Kinetic parameters for multiphase chemical reactions of OH with levoglucosan (white) and

abietic acid (gray) determined by the MCGA method of fitting the experimental data with the KM-GAP

model. The ranges of parameters are depicted as a box−whisker plot (the percentiles of 10, 25, 75, and 90%

are shown). Reprinted with permission from Arangio et al. (2015). Copyright 2015 American Chemical

Society.





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
