# Peer review of "Technical Note: Monte-Carlo genetic algorithm (MCGA) for"

_Atmospheric Chemistry and Physics, 2017_

## Referee Comment (RC1) · Anonymous Referee #1 · 9 Feb 2017

This technical note explains a method for inference of chemical and physical parameters relevant to atmospheric processes. As explained in the note, it could be of use to multiple types of atmospherically-relevant experiments investigating different parameters. It is therefore relevant to the journal and of importance to the research community.

I recommend publication pending minor revisions. Below are a description of these revisions (numbered). On the whole the note is very well written and presented, and I think goes into sufficient depth without being overbearing (as would be possible due to

the relatively complicated nature of the method in question).

1) The sentence spanning lines 103-106 is both complicated and elongated. Could it be made more readable?

2) In sect. 3 (around the genetic algorithm explanation) I am left unsure how homogenisation of the population is achieved. My interpretation of the text and Fig. 1 is that some set of the population with a satisfactorily high correlation survives and is not further changed. The remaining population of parameter sets (children) changes through recombination and mutation of extant children or through replacement of these children with new ones. How does this child population homogenise to a population with high correlation? Are their parameter values informed by the parent population (as the family names suggest)? If so, this needs to be made clearer I think. An alternative process that comes to mind is that the size of the parent population increases as more children meet the correlation criteria (i.e. a satisfactorily high correlation). They achieve this through the random process of parameter change (recombination etc) rather than through any inheritance from parents. Again, if this (or any other) process causes homogenisation then it needs to be explained more clearly in the text (and possibly in Fig. 1).

3) On lines 144-150 can some additional information be provided as to the relative pros and cons (if any) of the reseeding and migration approach vs. repetition of the MCGA approach? Furthermore, can statistical bounds be determined using the former approach as it is stated they can be for the latter?

4) In Fig. 1, I suggest making the distinction between the Monte-Carlo step and the genetic algorithm step clearer. From reading of the main text the difference is clear, however, the names of the two steps are combined in Fig. 1, and they could be separate and placed distinctly above their respective schematic representation. I only suggest this because it may make the concept of the approach easier to appreciate (I got confused with when the Monte-Carlo usage stopped due to the random nature of

mutations and introductions of new parameter sets in new generations of the genetic-algorithm step).

5) I ask the authors to consider expanding on their description of model development in the introduction to further emphasise the importance of the MCGA method. The increased model complexity they describe does allow for inference of parameter values from increasingly complex measurement setups. However, this is only possible through methods like MCGA. As atmospheric science tries to bridge the divide between laboratory measurements and the real atmosphere and simplified models and global ones, it seems that methods like the MCGA will be very important.

6) Typos: Should "similar model output" on line 27 be "similar model input"? Should "breath" on line 92 be "breadth"? Should "as heuristic" on line 110 read "as a heuristic"?

---

## Referee Comment (RC2) · Berkemeier et al. · 20 Feb 2017

The manuscript describes the principles of the Monte-Carlo genetic algorithm (MCGA)
and how it can be used to constrain various model input parameters for multiphase
chemical kinetic systems.

The manuscript is very well written and it is relatively straightforward to understand
the general idea, advantages and limitations of the MC genetic algorithm despite the
complex topic. I especially like the examples given in Figure 3 concerning why model
input parameter can remain unconstrained. I have very little additional to add apart

from what reviewer 1 already pointed out. I recommend the manuscript to be published after a minor revision where you consider the comments from reviewer 1, which I fully agree with, and my very minor additional comments given below.

On p. 2, L33-34: Do you really mean that the MCGA algorithm itself should be portable to any numerical model with similar computational expense and extent of the fitting parameter space or do you mean that the results (the constrained parameters) can be implemented in these models?

On p. 5, L87-90: This sentence is long and I had to read it several times before I understood the full meaning of it. Is it possible to reformulate it? Maybe: Furthermore, experiments covering a broad range of conditions must be conducted to achieve observables that are controlled by (a) as many model input parameters as possible across all experimental conditions, but (b) by as few model input parameters as possible for a specific experimental condition (i.e. limiting cases).

On p. 5, L92: I am not sure if I understand what you want to say with "in the required breath". Do you mean that because of technical limitations or transient behaviour it may not be possible to sample all required input parameters at the same time?

I agree with referee 1 that some additional information needs to be provided about the advantages of the reseeding and migration approach vs. repetition of the MCGA approach? Have you used the reseeding and migration approach for any of the results presented in the article? If I understand it correctly you used the repeated execution approach when you generated the results presented in Figure 3.

---

## Referee Comment (RC3) · Anonymous Referee #3 · 10 Mar 2017

The authors present a Monte-Carlo Genetic Algorithm tool for fitting large sets of input parameters of kinetic multiphase atmospheric chemistry box models using multiple experimental data sets. The manuscript is well written and is recommended for publication in ACP after the authors address the following minor comments.

1) Line 73: Please define the term "non-orthogonal input parameters".

2) Line 80-82: While I generally understand what the authors are trying to say here,

it would be useful to elaborate a bit on what the term "the most limiting processes" exactly means in this context. It would be great to briefly illustrate it with an example, if possible.

3) While MCGA will prove to be a powerful tool in interpreting experimental data, I appreciate the discussion of its limitations in section 3. This is not presently reflected in the abstract. I suggest adding a sentence in the abstract that cautions the future users of such a tool to its limitations as well as potential solutions to overcome them (e.g., broader range of experimental techniques and approaches, etc.).

—————————————————————

---

## Author Comment (AC1) · 9 May 2017

REFEREE #1

This technical note explains a method for inference of chemical and physical parameters relevant to atmospheric processes. As explained in the note, it could be of use to multiple types of atmospherically-relevant experiments investigating different parameters. It is therefore relevant to the journal and of importance to the research community.

I recommend publication pending minor revisions. Below are a description of these revisions (numbered). On the whole the note is very well written and presented, and I think goes into sufficient depth without being overbearing (as would be possible due to the relatively complicated nature of the method in question).

*We thank the reviewer for their effort and comments. The comments will be addressed individually, below.*

1) The sentence spanning lines 103-106 is both complicated and elongated. Could it be made more readable?

*The sentence has been adjusted. It originally read:*

*For cloud-aerosol interaction models, inverse modeling techniques using evolutionary algorithms as global optimization technique in Monte-Carlo Markov Chain (MCMC) algorithms were developed previously to determine parametric uncertainties.*

*it now reads:*

*A related technique (Monte-Carlo Markov Chain algorithm) has been used to determine parametric uncertainties in cloud-aerosol interaction models.*

2) In sect. 3 (around the genetic algorithm explanation) I am left unsure how homogenisation of the population is achieved. My interpretation of the text and Fig. 1 is that some set of the population with a satisfactorily high correlation survives and is not further changed. The remaining population of parameter sets (children) changes through recombination and mutation of extant children or through replacement of these children with new ones. How does this child population homogenise to a population with high correlation? Are their parameter values informed by the parent population (as the family names suggest)? If so, this needs to be made clearer I think.

*The reviewer correctly notes that 5 % of the individuals with the best goodness of fit survive and are passed directly to the next generation of the model as so-called elites (which we previously called parents), whereas the remaining 95 % of individuals of the next generation are created by recombination and mutation (80% vs. 20% respectively), this has*

been made clearer in the revised manuscript. Homogenization is achieved not only through survival of elites, but also through the parent selection process: the likeliness of contributing as a parent to a child for the next generation depends on the goodness-of-fit of the individuals. We have adjusted the statement in section 3 to make it clear how generations are formed. It originally read:

*The remaining population is generated using combinations of parameters from the individuals in the previous generation with moderate or better goodness-of-fit, forming the children for the next generation. To further ensure genetic variability, a mutation scheme alters parameters in a stochastic manner.*

It now reads:

*The remaining population is generated using combinations of parameters from the individuals in the previous generation with moderate or better goodness-of-fit (the parents), forming the children for the next generation. In this study, 5% of the next generation are elite individuals, which are transferred with no changes, while 80 % of the children is created by randomly choosing individual parameters (genes) from two selected parents with equal weighting. The higher the goodness-of-fit of a certain individual, the higher is its likeliness to be selected as parent. This way, parameters leading to high goodness-of-fit are positively reinforced, leading to improvement and slow homogenization of the population. Finally, 20 % of children are created by applying a mutation scheme that alters parameters in a stochastic manner within the prescribed bounds to enhance genetic variability.*

Figure 1 was modified to reflect these changes:

[Figure]

**Figure 1.** Schematic representation of the MCGA optimization method consisting of a Monte-Carlo sampling, which feeds into a genetic algorithm. *Populations* of model input parameter sets (blue boxes) are iteratively improved over several generations through survival of *elites* (red boxes) and recombination and mutation of *parents* to create *children*

(purple boxes), until a sufficient correlation to the experimental data (goodness of fit) is obtained.

An alternative process that comes to mind is that the size of the parent population increases as more children meet the correlation criteria (i.e. a satisfactorily high correlation). They achieve this through the random process of parameter change (recombination etc) rather than through any inheritance from parents. Again, if this (or any other) process causes homogenisation then it needs to be explained more clearly in the text (and possibly in Fig. 1).

This is an interesting idea to consider for future work. The current genetic algorithm operates with a fixed population size; this allows variables to be preallocated for computational efficiency.

3) On lines 144-150 can some additional information be provided as to the relative pros and cons (if any) of the reseeding and migration approach vs. repetition of the MCGA approach? Furthermore, can statistical bounds be determined using the former approach as it is stated they can be for the latter?

The reseeding / migration mechanisms were not used in this data presented in this manuscript and our informal tests indicate that it is not significantly faster to optimize using reseeding / migration than the repeated execution approach. We have removed the reference to reseeding and migration in the manuscript to reflect the work that we present and eliminate this point of confusion.

4) In Fig. 1, I suggest making the distinction between the Monte-Carlo step and the genetic algorithm step clearer. From reading of the main text the difference is clear, however, the names of the two steps are combined in Fig. 1, and they could be separate and placed distinctly above their respective schematic representation. I only suggest this because it may make the concept of the approach easier to appreciate (I got confused with when the Monte-Carlo usage stopped due to the random nature of mutations and introductions of new parameter sets in new generations of the genetic- algorithm step).

The figure has been adjusted according to the reviewer's suggestion.

5) I ask the authors to consider expanding on their description of model development in the introduction to further emphasise the importance of the MCGA method. The increased model complexity they describe does allow for inference of parameter values from increasingly complex measurement setups. However, this is only possible through methods like MCGA. As atmospheric science tries to bridge the divide between laboratory

measurements and the real atmosphere and simplified models and global ones, it seems that methods like the MCGA will be very important.

We are delighted that the reviewer finds the MCGA method worthwhile and wishes us to further highlight its utility. We have added the following statement to the end of the introduction (lines 101-104 of revised manuscript):

*The MCGA algorithm presented here is able to overcome the difficulty of a complex optimization hypersurface with many local minima while providing the user with a realistic assessment of how well-constrained the model input parameters are by the experimental data.*

6) Typos: Should "similar model output" on line 27 be "similar model input"?

The phrase should be "similar model output." There is some potential confusion between the kinetic model and the optimization algorithm. The MCGA algorithm optimizes "input parameters" while the kinetic model produces "output" for comparison to experimental data. We added the following paragraph to the introduction and hope this clarifies our nomenclature:

*We will use the term "input parameters" to address the prescribed model parameters (thermodynamic, kinetic, or physical) that are optimized in this study so that kinetic model output matches experimental data, a process that we will refer to as "fitting the kinetic model". Note that this definition excludes model parameters that are clearly defined by physical laws or the experiment (e.g. physical constants, experimental conditions) or are of purely technical nature (e.g. integration time steps).*

Should "breath" on line 92 be "breadth"?

Yes, the referee is correct. However, we decided that the sentence does not add much to the discussion at this point and removed it for simplification.

Should "as heuristic" on line 110 read "as a heuristic"?

Yes, it has been adjusted.

REFEREE #2

The manuscript describes the principles of the Monte-Carlo genetic algorithm (MCGA) and how it can be used to constrain various model input parameters for multiphase chemical kinetic systems.

The manuscript is very well written and it is relatively straightforward to understand the general idea, advantages and limitations of the MC genetic algorithm despite the complex topic. I especially like the examples given in Figure 3 concerning why model input parameter can remain unconstrained. I have very little additional to add apart from what reviewer 1 already pointed out. I recommend the manuscript to be published after a minor revision where you consider the comments from reviewer 1, which I fully agree with, and my very minor additional comments given below.

We thank the reviewer for their effort and comments. The specific comments will be addressed below.

On p. 2, L33-34: Do you really mean that the MCGA algorithm itself should be portable to any numerical model with similar computational expense and extent of the fitting parameter space or do you mean that the results (the constrained parameters) can be implemented in these models?

In this instance, we would like to express that MCGA can also be used with other numerical models. The abstract has been adjusted to reflect that we are using it for aerosol science, but that the method is portable to any process that can be numerically modeled. The constrained parameters generated by the MCGA should be portable to other models (e.g. of aerosol science), as long as the other model does not make other base assumptions.

On p. 5, L87-90: This sentence is long and I had to read it several times before I understood the full meaning of it. Is it possible to reformulate it? Maybe: Furthermore, experiments covering a broad range of conditions must be conducted to achieve observables that are controlled by (a) as many model input parameters as possible across all experimental conditions, but (b) by as few model input parameters as possible for a specific experimental condition (i.e. limiting cases).

The sentence has been adjusted using the suggestion of the reviewer with minor modification. It originally read:

*Furthermore, experiments must be conducted by covering broad ranges of experimental conditions to achieve that the observables are controlled by (a) as many model input*

*parameters as possible across all experimental conditions, but (b) by as few model input parameters as possible for a specific experimental condition (i.e. limiting cases).*

It now reads:

*Furthermore, experiments covering a broad range of conditions must be conducted to ensure that the observables are controlled by (a) as many model input parameters as possible across all experimental conditions, but (b) by as few model input parameters as possible for a specific experimental condition (i.e. limiting cases).*

On p. 5, L92: I am not sure if I understand what you want to say with "in the required breath". Do you mean that because of technical limitations or transient behaviour it may not be possible to sample all required input parameters at the same time?

We have adjusted the discussion of limitations on lines 90-95. What this statement meant is that it may not be physically possible to fully constrain all parameters, e.g., if a bulk reaction happens so "fast" that it is never the limiting process in experimental data, the corresponding parameter will have a lower limit set by experiment and an upper limit set by diffusion. However, we decided that the sentence does not add much to the discussion at this point and removed it for simplification.

I agree with referee 1 that some additional information needs to be provided about the advantages of the reseeding and migration approach vs. repetition of the MCGA approach? Have you used the reseeding and migration approach for any of the results presented in the article? If I understand it correctly you used the repeated execution approach when you generated the results presented in Figure 3.

Please see comments to reviewer 1. To answer the specific questions raised by reviewer 2: we have not used the reseeding and migration approach for the results in this article; as such, those statements have been removed from the article. The reviewer is correct in stating that the results in Figure 3 were generated via the repeated execution approach.

REFEREE #3

The authors present a Monte-Carlo Genetic Algorithm tool for fitting large sets of input parameters of kinetic multiphase atmospheric chemistry box models using multiple experimental data sets. The manuscript is well written and is recommended for publication in ACP after the authors address the following minor comments.

We thank the reviewer for their effort and comments. The specific comments will be addressed below.

1) Line 73: Please define the term "non-orthogonal input parameters".

We have changed the manuscript to include "(coupled)" following "non-orthogonal input parameters. A more thorough definition of the term "non-orthogonal parameters" is given in Sect. 3, line 193 and an example given on lines 207-213.

2) Line 80-82: While I generally understand what the authors are trying to say here, it would be useful to elaborate a bit on what the term "the most limiting processes" exactly means in this context. It would be great to briefly illustrate it with an example, if possible.

We have added more details and an example to the paper (lines 90-95 of revised manuscript). The text now reads:

*For example, if a model is trained using data that is exclusively limited by a single process, it will constrain the parameters that represent that specific process while the other parameters remain nearly unconstrained even if multiple data sets are used. This means that a parameter set were optimized using data from surface film experiments, the bulk diffusion coefficients would likely be poorly constrained regardless of how many different experimental datasets of that type were used.*

3) While MCGA will prove to be a powerful tool in interpreting experimental data, I appreciate the discussion of its limitations in section 3. This is not presently reflected in the abstract. I suggest adding a sentence in the abstract that cautions the future users of such a tool to its limitations as well as potential solutions to overcome them (e.g., broader range of experimental techniques and approaches, etc.).

We have adjusted the abstract to note that the MCGA is "allowing users to design experiments that should be particularly useful to constrain model parameters" (line 29-30 of revised manuscript).